# Effect of Low- and High-Sulfur-Containing Amino Acid Inclusion in Diets Fed to Primiparous Sows in Late Gestation on Pre-Partum Nitrogen Retention and Offspring Pre- and Post-Weaning Growth Performance

**DOI:** 10.3390/ani14243681

**Published:** 2024-12-20

**Authors:** Cristhiam Jhoseph Munoz Alfonso, Lee-Anne Huber, Crystal L. Levesque

**Affiliations:** 1Department of Animal Biosciences, University of Guelph, Guelph, ON N1G 2W1, Canada; munozalc@uoguelph.ca (C.J.M.A.); huberl@uoguelph.ca (L.-A.H.); 2Department of Animal Science, South Dakota State University, Brookings, SD 57006, USA

**Keywords:** sulfur-containing AAs, primiparous sows, offspring performance, gestation

## Abstract

This study investigated the impact of low and high levels of sulfur-containing amino acids (SAAs) in the diets of pregnant sows on pre-partum nitrogen retention and subsequent growth and plasma sulfur-containing metabolites such homocysteine (Hcys), cysteinyl–glycine, glutathione (GSH), and taurine (Tau) of the offspring. Higher SAA levels in the sows’ diet led to increased nitrogen retention in late gestation and did not translate to improved growth performance of the piglets in the suckling or post-weaning phases. Interestingly, this study also revealed that the sow can regulate the transfer of certain nutrients, like taurine (Tau), to the litter, ensuring their supply even when the maternal diet is not optimal. This research highlights the complex relationship between maternal nutrition and offspring development, suggesting that while adequate SAA intake is important for the sow, excessive levels may not offer additional benefits to the sow or piglets in terms of growth and production of metabolites such as antioxidants that are important for the health status of the animals. These findings emphasize the need for carefully balanced diets for pregnant sows to support both maternal growth and nitrogen retention and the potential impact for the offspring in the synthesis of biomolecules that have been associated with better health status.

## 1. Introduction

The negative impacts of a suboptimal content of amino acids (AAs) in a diet fed to sows during gestation have been previously demonstrated [1]. For example, in pregnant sows, the inadequate intake of indispensable and dispensable AAs can reduce fetal development, with long-term repercussions for post-natal growth performance [2], and the performance of the sows. Primiparous sows could be even more susceptible than mature sows to inadequate dietary AA supply, since nutrients are needed both for supporting protein deposition in the product of conceptus and for maternal growth [3]. Failing to achieve the optimal inclusion of essential AAs in the diet forces the primiparous sow to mobilize maternal nutrient reserves and adjust the growth of maternal tissues. Meanwhile, a severe restrictive condition may also lead to poor fetal development, resulting in perturbed offspring performance during the nursery, growing, and finishing stages [4]. This poor development is not only related to restricted protein synthesis in the fetal body but also a potential reduction in the synthesis of biomolecules from both the sows and the offspring that correspond to the non-protein uses of AAs. The use of AAs for the synthesis of non-protein molecules has not yet been included in the requirement estimates for gestating primiparous sows, despite using a large portion of available AAs in some cases [5,6]. Consequently, it is pertinent to study the effect of dietary AA levels fed during late gestation on non-protein biomolecule synthesis, whole-body protein retention, and reproductive performance simultaneously. In particular, sulfur-containing AAs (SAAs) are of interest because of their participation in the synthesis of biomolecules such as creatine, phosphatidylcholine, glutathione (GSH), and taurine (Tau) needed for the growth and development of both primiparous sows and piglets [7]. It has been hypothesized that the inclusion of SAAs may influence all AAs for protein and non-protein uses. Therefore, the objective of this study was to examine the inclusion of both low and high dietary SAAs in diets fed to primiparous sows during late gestation to elucidate the impact on pre-partum nitrogen retention and offspring pre- and post-weaning growth performance.

## 2. Materials and Methods

This study was approved by the Institutional Animal Care and Use Committee at the South Dakota State University and Use Committee (IACUC #2205-024A).

### 2.1. Animals and Housing

A total of 30 primiparous sows (PIC 1050) were included in this study. Primiparous sows were bred at day 470 ± 5 of life, and on day 89 ± 1 of gestation, they were selected from the high health herd of the South Dakota State University Swine Education and Research facility and randomly allotted to 1 of 2 dietary treatments (0.29% and 0.92% standardized ileal digestible (SID) SAAs). Water was offered ad libitum. On day 110 of gestation, primiparous sows were moved to farrowing crates. Piglets were cross-fostered within the first 24 h of parturition within the treatments to achieve one piglet per functional teat, followed by standard litter processing procedures. Creep feed was not supplied to the experimental crates. At weaning on day 23 ± 5, litters from sows receiving the dietary treatments remained together (up to 8 pigs/litter), placing 4 females and 4 males per pen, for six weeks.

### 2.2. Diets and Feeding

Two experimental diets were formulated using corn, peas, rye, soybean hulls, cornstarch, and feather meal to achieve 63 or 200% of the estimated SID SAAs requirement for primiparous sows in late gestation (0.46% SID SAAs; requirements from d 90 to d 114 of gestation, 145 kg of body weight at breeding, expected litter size of 15, and piglet birth weight of 1.35 kg) using the expected parameters of the breed [8] and historical barn data, which correspond to 0.29% and 0.92% SID SAAs, respectively (Table 1). Diets were provided in mash form. In order to achieve the highest SID SAAs inclusion, feather meal and peas were included in the diets, along with crystalline cysteine (Cys). All indispensable AAs except methionine (Met) were provided at or above the estimated requirements for primiparous sows during late gestation [8], and the diets were isoenergetic. Titanium dioxide was included in the diets at 0.13 and 0.14% as an indigestible marker to determine the apparent total tract N digestibility. The experimental diets (2.50 kg per day) were fed between d 90 of gestation and farrowing in two equal meals. After farrowing, all sows received a common lactation diet (Table 2) using a step-up feeding program to achieve ad libitum intake within five days after farrowing. All weaned pigs were provided common nursery diets in a four-phase feeding program (Phase I: wean to d 7; Phase II: d 8 to d 21; Phase III: d 22 to d 35; Phase IV: d 36 to end of trial; Appendix A).

### 2.3. Sample and Data Collection

Growth performance: Sows were weighed and their backfat was measured by ultrasound on day 89 of gestation, within 24 h of farrowing, and at weaning (23 ± 5 d of lactation). Individual piglets were weighed within 18h of birth and at weaning. After weaning, individual pig weight and per-pen feed disappearance were recorded for growth performance evaluation on d7, d14, d21, d28, d35, and d42.

Nitrogen (N) balance: Between d 107 and d 109, an N-balance study was performed using total urine collection via urinary catheters (Lubricath^®^, 2-way, 30 cc balloon, 18 FR, Bard Medical Canada Inc., Oakville, ON, Canada) and daily fecal grab sampling as described by Huber et al. [9]. To avoid bacterial growth and volatilization of N, 15 mL of sulfuric acid was added daily to the urine collection containers to maintain a pH below 3. A 5% (*w*/*w*) urine subsample was collected daily, pooled per gilt, and stored at −20 °C until further analysis.

Blood collection: Ten milliliters of blood was taken from all the fasted primiparous sows at the beginning of this study (d90), within 24 h after farrowing, and at weaning from the orbital sinus using a 14 ga × 5.1 cm needle and heparinized blood collection tubes (Fisher Scientific, Hampton, NH, USA), utilizing a snare to restrain them. Immediately after birth before suckling and at weaning, blood (5 mL) was collected from two piglets per litter by jugular venipuncture (20 ga × 2.5 cm needle) into a heparinized vacutainer (Fisher Scientific, Hampton, NH); piglets with BW at birth close to the overall mean (1.30 kg) and an approximately equal number of females and males per treatment (8 males and 7 females from 0.29% SID SAA-fed sows and 6 males and 6 females from 0.92% SID SAA-fed sows) were selected for the blood collection. During the nursery period, blood samples were collected from 2 pigs per pen on d21 and d42, maintaining an equal number of females and males per treatment and using the same selection criteria. Therefore, the same piglets may not have been sampled multiple times. Feeders were removed from the pens the evening before to guarantee the post-absorptive stage in the animals. All blood was separated by centrifugation at 3500× *g* for 15 min, and plasma was removed and stored at −70 °C until concentrations of plasma AAs and biomolecules of methionine (Met) metabolism were quantified.

### 2.4. Chemical Analyses

The experimental diets were analyzed for CP by calculating N × 6.25 and N measured using the combustion procedure (method 990.03; [10] Int., 2019; University of Missouri, Columbia, MO, USA). Crude fat (Ether Extraction, [10] Official Method 920.39), crude fiber ([10], Official Method 978.10, 2006), neutral detergent fiber (NDF; [10] 56, 1352–1356, 1973), and acid detergent fiber (ADF; [10] Official Method 973.18 (A–D), 2006; University of Missouri, Columbia, MO, USA) were also assessed. Gross energy (GE) was analyzed using an isoperibol bomb calorimeter and benzoic acid as the standard for calibration (Parr 6300 calorimeter, Parr Instruments Co., Moline, IL, USA). Urine and fecal samples were analyzed for N contents applying the same procedure described previously, using LECO FP628 (LECO CORPORATION, Saint Joseph, MI, USA; University of Guelph, Guelph, On, Canada). The titanium content in the experimental diets and fecal samples was analyzed following the procedure described by Christensen and Huber [11] to determine the apparent total digestibility (ATTD) of N.

Dietary amino acids were analyzed on a Hitachi Amino Acid Analyzer, Model No. L8800 (Hitachi High Technologies America, Inc., Pleasanton, CA, USA) using ninhydrin for postcolumn derivatization and norleucine as the internal standard. Prior to analysis, samples were hydrolyzed with 6N HCl for 24 h at 110 °C (method 982.30 E(a); [10] Int., 2019). Methionine and cysteine (Cys) were determined as met sulfone and cysteic acid, respectively, after cold performic acid oxidation overnight before hydrolysis (method 982.30 E(b); [10] Int., 2019). Tryptophan was determined after NaOH hydrolysis for 22 h at 110 °C (method 982.30 E(c); [10] Int., 2019).

Blood plasma samples were analyzed for concentrations of dispensable and indispensable AAs, homocysteine (Hcys), Tau, and glutathione (GSH) using ultra-performance liquid chromatography [12]. All AAs were analyzed using 10% sulfosalicylic acid as the deproteinizer and norvaline as the internal standard according to the method of Boogers et al. [13], and using Ultra-Performance Liquid Chromatography and Empower Chromatography Data Software version 3 (Waters Corporation, Milford, MA). Total plasma Hcys, cysteinyl–glycine, and GSH were measured according to previously published methods [14,15], with minor adaptations, as described by Banton et al. [12]. The derivatized thiols were separated using an Acquity UPLC BEH C18 column (2.1 × 50 mm, 1.7 μm; Waters Corporation, Milford, MA, USA) maintained at 28 °C with fluorescence detection at 515 nm emission and 385 nm excitation. The peak areas obtained were compared with known standards and analyzed using the Waters Empower 2 Software (Waters Corporation, Milford, MA, USA).

### 2.5. Calculations and Statistical Analysis

The ATTD of N was calculated by the index method [16]. Total N retention in the body was determined as per the method outlined previously [17]. Nitrogen intake during late gestation was computed by multiplying the analyzed N content in the diet with the corresponding feed allowance. Feed refusal was not a significant factor during the experiment; instances of incomplete feed consumption were excluded from the nitrogen balance assessments, and due to the singular diet batch utilized, each treatment was assigned a unique nitrogen intake value, impeding the calculation of the standard error. Equality of variances among the treatments was tested using the Folded F method of SAS^®^ software (version 9, SAS Inst. Inc., Cary, NC, USA). Data were then analyzed using the *t*-test procedure with the main effect of dietary treatment for N utilization and the litter and post-weaning performance of offspring. A two-way ANOVA test was performed separately to assess the main effects of dietary treatment, stage (at farrowing and at weaning for sows; at farrowing, at weaning, and 3 and 6 weeks post-weaning for piglets), and diet × stage interaction in repeated measurements for plasma concentrations of AAs and Met metabolites. Contrast statements were used to determine the differences between treatments within the stage when the diet × stage interaction was statistically significant. Statistical significance and tendencies were stated at *p* < 0.05 and 0.05 ≤ *p* < 0.10, respectively.

## 3. Results

Two primiparous sows were excluded due to the low intake of the experimental diet, and a third one was excluded due to an abortion, all of which were from the dietary treatment containing 0.92% SID Met. The data were not included in the statistical analyses.

### 3.1. Experimental Diets

The analyzed crude protein contents correspond with the calculated values for the experimental diets (Table 1 and Table 2). The analyzed values of Lys and Met were consistent with the calculated values (Table 1 and Table 2). The Cys content, however, was 15 and 47% greater than that calculated in the 0.29 and 0.92% SID SAA diets, respectively. This discrepancy might be attributed to an underestimation of total Cys derived from feather meal during diet formulation. Nevertheless, when considering all the analyzed and calculated values for total SAAs, the distinction between treatments was sufficient to fulfill the objectives of this study.

### 3.2. Nitrogen Balance

Total, urinary, and fecal N excretion, as well as ATTD of N and whole-body N retention (g/d), were greater (*p* < 0.05) for primiparous sows fed the diet containing 0.92% versus 0.29% SID SAAs (Table 3). Whole-body N retention as a percent of intake did not differ between treatment groups.

### 3.3. Pre- and Post-Weaning Performance

Litter characteristics at birth (number born alive, stillborn, mummies, litter size after standardization) and the subsequent lactation performance of the primiparous sows (sow body weight, backfat thickness change, feed intake) or litters (piglet body weight at birth or at weaning or average daily gain) were not influenced by dietary SID SAAs in late gestation (Table 4). In addition, the post-weaning growth performance of the offspring was not influenced by maternal dietary treatment in late gestation (Table 5).

### 3.4. Post-Absorptive Plasma Sulfur-Containing Metabolites

Primiparous sow plasma concentrations of Hcys, cysteinyl–glycine, GSH, Met, and Cys were not influenced by the main effect of treatment or the interaction between treatment and stage. Plasma concentrations of Hcys, cysteinyl–glycine, and GSH decreased between farrowing and weaning regardless of dietary treatment (*p* < 0.001, *p* < 0.005, and *p* = 0.098, respectively). Plasma Tau was influenced by the interaction between dietary treatment and stage, as well as the main effects of diet and stage (*p* = 0.001, 0.003, and <0.001, respectively; Table 6), where plasma Tau was greater (contrast statement; *p* < 0.001) at farrowing in primiparous sows fed 0.92% SID SAAs and not different at weaning compared to sows fed 0.29% SID SAAs. For the offspring, plasma Hcys was influenced by the interaction between (maternal) diet and stage (*p* < 0.001; Table 7), where piglets from primiparous sows fed 0.92% SID SAAs had greater plasma Hcys (contrast statement; *p* < 0.001) at weaning but were not different at birth or at 3 and 6 weeks after weaning compared to offspring from 0.29% SID SAA-fed sows. Plasma Hcys also tended to be greater for offspring of primiparous sows that received 0.92% SID AAs (*p* = 0.066; Table 7), but there were no other (maternal) diet effects on plasma concentrations of sulfur-containing metabolites in the offspring. Conversely, the main effect of stage influenced most sulfur-containing metabolites except for plasma GSH (*p* = 0.105). There was a significant increase in Hcys from birth to weaning, followed by a slight decrease and stabilization. A similar pattern was observed for cysteinyl–glycine, with a significant increase from birth to weaning, and then a decrease and stabilization. And for Met, there was a significant increase from birth to weaning and then a slight decrease and stabilization. In contrast, Cys had a significant decrease from birth to weaning, followed by a slight decrease and then increase. Taurine, on the other hand, was maintained from birth to weaning, followed by a decrease and then increase by 6 weeks post-weaning.

### 3.5. Post-Absorptive Plasma AA Profile

Primiparous sow plasma concentrations of Val and Ser were influenced by the interaction between dietary treatment and stage, as well as the main effects of diet and stage (*p* = 0.075, 0.018, and <0.001 for Val, and *p* = 0.040, 0.035, and 0.071 for Ser, respectively; Table 8), where plasma Val and Ser were greater (contrast statements; *p* = 0.003 and *p* = 0.006, respectively) at farrowing in primiparous sows fed 0.92% SID SAAs and not different at weaning compared to sows fed 0.29% SID SAAs. The post-absorptive plasma concentrations of Ile, Leu, Val, Tyr, and Ser were greater for primiparous sows fed 0.92 versus 0.29% SID SAAs (main effect of dietary treatment; *p* = 0.049, 0.042, 0.018, 0.039, and 0.071, respectively). The post-absorptive plasma concentrations of most dispensable and indispensable AAs decreased between farrowing and weaning (main effect of stage; *p* < 0.05), except for Ile, Leu, Met, Trp, and Asp, which were not affected by stage. The post-absorptive plasma concentrations of Val and Ser increased between farrowing and weaning (*p* < 0.01). For the offspring, there were no diet or interactive effects for any of the dispensable and indispensable AA concentrations in plasma (Table 9), though most AAs, except for His, Gly, and Pro, were influenced by the main effect of the stage. The majority of indispensable AAs generally show an increasing trend from birth to weaning, followed by a plateau or slight decrease, except for Lys, which decreased then increased. Moreover, total indispensable AAs seem to increase as stage changes. In contrast, dispensable AAs show a decreasing trend from birth to weaning, followed by a stabilization or slight increase, except for Asn, which increased by 3 weeks post weaning, while Gly and Tau decreased by 3 weeks post weaning.

## 4. Discussion

The purpose of the present study was to examine the inclusion of low and high dietary SAAs in diets fed to primiparous sows during late gestation to elucidate effects on pre-partum nitrogen retention and offspring pre- and post-weaning growth performance. The temporal concentrations of Met metabolites in the plasma were also assessed to infer changes in Met metabolism in both the sow and the offspring as it related to the SAA feeding level in late gestation. The plasma Met metabolite concentrations were also used as a tool to enrich our knowledge about the requirement of SAAs for non-protein uses, such as the synthesis of biomolecules via the transsulfuration pathway, including the antioxidants Tau and GSH.

The N balance study demonstrated greater whole-body N retention when sows received 0.92% SID inclusion of SAAs compared with sows receiving 0.29% SID SAAs, confirming that the 0.29% SID SAAs diet was limiting in SAAs for whole-body protein deposition. Although the retention reported in the current study is in agreement with previous studies [17,18], the improvement in N retention was accompanied by greater N excretion, indicating that the optimal dietary SAA inclusion was surpassed despite greater protein deposition in the whole body, which includes maternal and fetal tissues as well as pregnancy-associated tissues such as the placenta, other membranes, and the mammary gland [19]. It is possible that an AA other than Met became limiting for protein deposition in the sows that received the 0.92% SID SAAs diet, specifically Lys or Thr due to their numerically lower concentration in plasma for the 0.92 vs. the 0.29 group at farrowing. Moreover, there were no differences in sow body weight after farrowing or litter birth weight, indicating that additional protein retention occurred, but it was not detected in the body weights of the sows or offspring. Indeed, the additional N retained by sows that received the 0.92% SID SAAs diet can be translated into a maximum of 1.3 kg of protein at the end of the 26 d feeding period, making it challenging to determine which protein pool benefited. Likewise, no differences in the litter growth rate during the subsequent lactation period were found, suggesting no improvement in the milk production potential of the sow. Therefore, providing SAAs 200% above the estimated requirements appears to more than meet the needs of the primiparous sow for supporting protein deposition during late gestation.

The post-weaning performance of the offspring from sows receiving the experimental diets during late gestation did not show any differences in the current study. This was a rather predictable outcome as neither the litter characteristics nor piglet growth during the suckling period were affected by dietary treatment, although the lack of (maternal) dietary treatment effects on piglet growth rates continued throughout nursery, based on the reported impact of the nutrient density of maternal diet on fetal development [20], not only on fetal growth in utero but also on metabolic health and disease susceptibility later in life [21,22,23]. In the current study, it has been demonstrated that the primiparous sow had the ability to buffer dietary AA imbalances, safeguarding fetal development and further supporting the lack of difference in offspring growth performance through the end of the nursery period. This effect was also described by Thayer et al. [24], who proposed the Maternal Nutritional Buffering Model, which suggests that short-term changes in maternal macronutrient intake during pregnancy have minimal effects on fetal development due to internal buffering mechanisms. However, the model mentioned above contrasts with essential micronutrients, which must be sourced from diet and are more directly affected by maternal intake.

Our study also aimed to describe the effect of the under- and over-supply of SAAs on the production of biomolecules through the transsulfuration pathway, such as Tau and GSH. In this work, the plasma concentration of Tau in sows at farrowing was greater when the dietary SAA supply in late gestation was elevated. Beyond its well-known antioxidant role, Tau is crucial for various functions in the body, and during pregnancy, Tau supports the growth and brain development of the fetuses and must be provided by the mother during gestation and during the first week of lactation to piglets due to limited neonatal enzyme activity [25]. Even though circulating Tau makes up about 30% of the whole body, as described by Lambert et al. [26], this may reflect the whole-body Tau status of the animal. In addition, it was speculated that the greater plasma concentration of Tau in sows prior to farrowing may be reflected in the greater plasma concentration of Tau in piglets at birth. This was not the case, however, bringing about the hypothesis that there must be a tightly controlled maternal–fetal Tau transfer mechanism during the peripartum period, and the majority of this supply to the offspring may come during the first week of lactation, as illustrated in the study of Zaima [27]. Also, the reduction in the concentration of plasma metabolites such Hcys and cysteinyl–glycine from farrowing to weaning in sows, independent of dietary treatment during late gestation, may be evidence of the active transference of biomolecules synthesized via the transsulfuration pathway through lactation in favor of offspring development.

The effect of dietary SAAs fed to sows during late gestation in the offspring at weaning was evident for Hcys, indicating possible alterations in the transsulfuration or remethylation pathways. Plasma Hcys concentrations are highly regulated by cystathionine β-synthase and cystathionine γ-lyase [28] when transsulfuration is occurring, as well as methionine synthase and betaine-homocysteine methyl transferase [29] when remethylation occurs. Others have conducted carry-over studies and found increased enzyme activities and concentrations of sulfur-containing metabolites [30,31] in mothers and offspring when Met was supplemented. However, the greater weaning plasma Hcys concentration in the descendants of sows fed 0.92% SID SAAs during late gestation may suggest that the enzyme activities present in the transsulfuration and remethylation pathways were reduced, resulting in an accumulation of Hcys in offspring plasma. Presumably, the Hcys in offspring plasma during the neonatal period must be transferred from the mother, as it has been shown that the enzyme activities of human offspring involved in transsulfuration are low or absent during the neonatal period, depending totally on the mother’s transference [32]. Nevertheless, it seemed to be a short-term effect as most of the sulfur-containing metabolites were not different between the two offspring groups throughout the nursery period.

Previous studies have shown that the dietary levels of SAAs can have a significant impact on the plasma AA profile [33,34]. In this study, the increase in the plasma concentrations of Leu, Val, Ser, and Tyr when SAAs were provided above the current NRC recommendation is considered a positive effect, as there was no negative impact on the N balance or sow performance, presumably making these AAs available for non-protein uses in the body. Of particular interest is Ser, which is produced endogenously and participates in the synthesis of sphingolipids, essential components of cellular membranes, and plays important signaling functions in various physiological processes [35]. Ser may be considered a conditional essential AA as it is utilized for the synthesis of Cys and Gly, the antioxidants Tau and GSH in the transsulfuration pathway, and serves in protein polarization with therapeutic uses in the body [36]. When Ser is metabolized, it can also be used as an energy source or for lipid synthesis. However, Ser can decrease Tau in the brain as it is a neuro modulator [37], but no other up-regulation has been described so far in any other study. In addition, extra circulating Leu can also boost the utilization of AAs by the mammary gland as it promotes the activity of the mammalian target of rapamycin, which can lead to an improvement in milk production [38]. Nevertheless, the concentrations of these AAs in sow plasma were not different by weaning, which means that the diet-induced impact on the maternal AA metabolism is transient and disappears after a ~22-day period, where all sows received the same lactation diet. Moreover, the plasma AA concentrations of the offspring were not affected at any measured time point by the dietary treatment provided to their progenitor during late gestation, except for Val. One more time, the buffer capacity of gestating sows was evidenced when dietary AA imbalances are present, and capacity persists even when specific AA intake deviates significantly from the estimated requirements and notably extends to primiparous sows undergoing concurrent maternal growth. This observation emphasizes the prioritization of the sows for fetal AA supply, even in the face of increased maternal–fetal competition for these critical nutrients.

## 5. Conclusions

The most notable observation is the sow’s remarkable ability to buffer dietary AA imbalances, particularly during late gestation. This ensures that fetal development remains unaffected, even when the maternal intake of SAAs is substantially above or below the recommended levels, especially for primiparous sows. The positive impact on whole-body N retention did not translate to notable differences in offspring growth performance when dietary SAAs were provided above the estimated NRC requirements. However, it was evident that metabolites resulting from the metabolism of SAAs can be influenced in the peripartum stage from maternal dietary treatment; moreover, these effects can be seen to be translated into the offspring, if in the short-term, demonstrating the non-protein uses of SAAs.

## Figures and Tables

**Table 1 animals-14-03681-t001:** Ingredient composition and nutrient contents of experimental diets (as-fed basis) ^1^.

Item	Dietary Treatment ^1^	
	0.29	0.92
Ingredient composition, %		
Corn	19.55	20.46
Field peas	30.00	20.50
Rye	16.21	16.96
Soybean hulls	12.20	12.76
Cornstarch	11.06	11.58
Feather meal	3.30	9.50
Soybean oil	2.37	2.48
Vitamin and mineral premix ^2^	0.25	0.25
Calcium phosphate	1.23	1.25
Limestone	1.35	1.31
Salt	0.32	0.34
L-Lys-HCl	0.43	0.50
DL-Met	0.00	0.33
L-Thr	0.24	0.16
L-Trp	0.10	0.10
L-Val	0.12	0.12
L-Cys	0.00	0.12
L-His	0.02	0.03
L-Ile	0.09	0.10
L-Leu	0.06	0.00
L-Glu	0.97	1.01
Titanium dioxide	0.13	0.14
Total	100.00	100.00
Calculated nutrient contents		
Crude protein, %	15.63	18.97
Net energy, kcal/kg	2465	2438
Calcium, %	0.87	0.88
Total P, %	0.54	0.52
Standardized total tract digestible P,%	0.38	0.38
Total Lys, % ^3^	1.10 (0.80)	1.13 (0.80)
Total SAAs, %	0.49 (0.29)	1.08 (0.92)
Total Met, %	0.16 (0.07)	0.51 (0.40)
Total Cys, %	0.33 (0.22)	0.57 (0.52)

^1^ Experimental diets were fed from day 90 of gestation and farrowing and provided 63 or 200% of the estimated standardized ileal digestible sulfur AA requirements (NRC, 2012). Lactation diet was fed during a 23 ± 5-day period. Dietary treatments were as follows: 0.29% and 0.92% standardized ileal digestible sulfur AAs, which correspond to 63 or 200% of the estimated SID SAAs requirement for primiparous sows in late gestation (requirements from d 90 to d 114 of gestation, 145 kg of body weight at breeding, expected litter size of 15, and piglet birth weight of 1.35 kg). ^2^ Provided per kg of premix: vitamin A, 22,046,000 IU; vitamin D3, 4,188,740 IU; vitamin E, 189,992 IU; vitamin K, 8818 mg as menadione; pantothenic acid, 66,138 mg; riboflavin, 198,414 mg; folic acid, 8836 mg; niacin, 110,472 mg; thiamin, 6613 mg; pyridoxine, 30,313 mg; vitamin B_12_, 88,184 µg; biotin, 795 mg; Cu, 8088 ppm; Fe, 80,882 ppm; Mn, 21,617 ppm; Zn, 80,882 ppm, as Zinc 80,882 ppm; Se, 147.05 ppm; I, 176.5 ppm. ^3^ Standardized ileal digestible values are provided in parentheses.

**Table 2 animals-14-03681-t002:** Analyzed nutrient contents of experimental diets and standard lactation diet (as-fed basis) ^1^.

Item	Dietary Treatment ^1^		Standard Lactation Diet
	0.29	0.92	
Crude protein, %	14.83	19.69	18.38
Gross energy, Kcal/kg	3838	3943	-
Crude fat	2.02	2.42	2.45
Crude fiber	6.38	6.49	2.42
Ash	4.36	4.51	5.15
NDF	13.15	14.47	9.92
ADF	11.00	13.74	6.09
Indispensable amino acids, %			
Arg	0.86	1.07	0.96
His	0.31	0.32	0.47
Ile	0.63	0.90	0.74
Leu	1.11	1.38	1.66
Lys	1.04	1.02	1.21
Met	0.15	0.41	0.30
Phe	0.66	0.86	0.87
Thr	0.39	0.43	0.54
Trp	0.14	0.16	0.19
Val	0.84	1.31	0.90
Dispensable amino acids, %			
Ala	0.65	0.83	0.99
Asp	1.24	1.42	1.57
Cys	0.38	0.84	0.31
Glu	2.95	2.88	3.04
Gly	0.75	1.10	0.71
Pro	0.95	1.45	1.16
Ser	0.72	1.18	0.69

^1^ Experimental diets were fed to sows between gestation day 90 and farrowing and provided 63 or 200% of estimated standardized ileal digestible sulfur AA requirements (NRC,2012). Lactation diet was fed during a 23 ± 5-day period. Dietary treatments were as follows: 0.29% and 0.92% standardized ileal digestible sulfur AAs, which correspond to 63 or 200% of the estimated SID SAAs requirement for primiparous sows in late gestation (requirements from d 90 to d 114 of gestation, 145 kg of body weight at breeding, expected litter size of 15, and piglet birth weight of 1.35 kg).

**Table 3 animals-14-03681-t003:** Nitrogen utilization in gestating primiparous sows fed diets containing 0.29% or 0.92% standardized ileal digestible sulfur amino acids between d 107 and 109 of gestation.

Item	Dietary Treatment ^1^		SEM ^2^	*p*-Value ^3^
	0.29	0.92		
No. of sows	15	12	-	*-*
N intake, g/d	59.25	78.75	-	-
Total N excretion, g/d	40.00	51.51	1.826	<0.001
Urinary N excretion, g/d	16.24	24.79	1.904	<0.001
Fecal N excretion, g/d	23.76	26.72	0.783	0.001
Total urine excretion, kg/d	3.72	5.26	1.55	0.331
ATTD of N, % ^4^	59.91	66.07	1.228	<0.001
Whole-body N retention, g/d	19.25	27.24	1.826	<0.001
Whole-body N retention, %	32.50	34.59	2.653	0.438

^1^ Experimental diets were fed to sows between gestation day 90 and farrowing. Dietary treatments were as follows: 0.29% and 0.92% standardized ileal digestible sulfur AAs, which correspond to 63 or 200% of the estimated SID SAAs requirement for primiparous sows in late gestation (requirements from d 90 to d 114 of gestation, 145 kg of body weight at breeding, expected litter size of 15, and piglet birth weight of 1.35 kg). ^2^ Maximum value for the standard error of the means. ^3^ *p*-values for the *t*-test procedure between treatments. ^4^ Apparent total tract digestibility of nitrogen.

**Table 4 animals-14-03681-t004:** Subsequent lactation performance of primiparous sows fed diets containing 0.29% or 0.92% standardized ileal digestible sulfur amino acids between d90 of gestation and farrowing.

Item	Dietary Treatment ^1^		SEM ^2^	*p*-Value ^3^
	0.29	0.92		
No. of sows	15	12	-	-
Sow body weight at farrowing, kg	197.9	193.7	3.34	0.221
Sow body weight at weaning, kg	177.1	174.0	4.36	0.488
Sow body weight change, kg ^4^	−20.8	−19.7	3.09	0.717
Sow average daily feed intake, kg	4.85	4.66	0.22	0.406
Sow backfat change from late gestation to farrowing ^4^	−4.13	−3.33	0.91	0.470
Sow backfat change from farrowing to weaning ^4^	−1.20	−1.08	0.87	0.911
Piglets				
Born alive, no.	14.0	13.9	1.05	0.942
Stillborn, no.	1.33	2.00	0.71	0.357
Mummies, no.	0.20	0.69	0.31	0.122
Litter size after standardization	12.7	13.0	0.93	0.709
Litter size at weaning	11.1	10.9	0.94	0.821
Body weight at birth, kg	1.49	1.44	0.08	0.476
Body weight at weaning, kg	6.79	6.64	0.38	0.689
Litter overall average daily gain, kg	0.260	0.266	0.02	0.720

^1^ Dietary treatments were as follows: 0.29% and 0.92% standardized ileal digestible sulfur AAs, which correspond to 63 or 200% of the estimated SID SAAs requirement for primiparous sows in late gestation (requirements from d 90 to d 114 of gestation, 145 kg of body weight at breeding, expected litter size of 15, and piglet birth weight of 1.35 kg). ^2^ Maximum value for the standard error of the means. ^3^ *p*-values for the *t*-test procedure between the treatments. ^4^ Differences refer to the final vs. initial values.

**Table 5 animals-14-03681-t005:** Post-weaning growth performance of offspring from primiparous sows fed diets containing 0.29% or 0.92% standardized ileal digestible sulfur amino acids between d90 of gestation and farrowing.

Item	Dietary Treatment ^1^		SEM ^2^	*p*-Value ^3^
	0.29	0.92		
Wean BW ^4^, kg	6.63	6.62	0.397	0.982
Phase I				
FBW, kg	7.74	7.60	0.520	0.778
ADFI, kg	0.16	0.15	0.021	0.489
ADG, kg	0.14	0.15	0.027	0.938
G:F	0.85	0.93	0.112	0.499
Phase II				
FBW, kg	13.07	12.47	0.777	0.450
ADFI, kg	0.47	0.44	0.035	0.379
ADG, kg	0.38	0.35	0.025	0.216
G:F	0.81	0.80	0.037	0.581
Phase III				
FBW, kg	22.10	21.04	1.197	0.386
ADFI, kg	0.80	0.74	0.039	0.115
ADG, kg	0.65	0.61	0.033	0.308
G:F	0.81	0.83	0.021	0.366
Phase IV				
FBW, kg	28.18	27.12	1.143	0.364
ADFI, kg	1.23	1.23	0.048	0.922
ADG, kg	0.85	0.86	0.038	0.871
G:F	0.70	0.71	0.040	0.772
Overall				
ADFI, kg	0.66	0.62	0.030	0.258
ADG, kg	0.51	0.49	0.021	0.261
G:F	0.78	0.79	0.013	0.714

^1^ Dietary treatments were as follows: 0.29% and 0.92% standardized ileal digestible sulfur AAs, which correspond to 63 or 200% of the estimated SID SAAs requirement for primiparous sows in late gestation (requirements from d 90 to d 114 of gestation, 145 kg of body weight at breeding, expected litter size of 15, and piglet birth weight of 1.35 kg). ^2^ Maximum value for the standard error of the means. ^3^ *p*-values for the *t*-test procedure between maternal treatments. ^4^ Pigs were weaned by sows into pens, with 8 pigs per litter per pen. Common diets were provided in a 4-phase feeding program (Phase I: wean to d 7; Phase II: d 8 to d 21; Phase III: d 22 to d 35; Phase IV: d 36 to end of trial). BW = body weight, FBW = final body weight ADFI = average daily feed disappearance, ADG = average daily gain, G:F = gain-to-feed ratio.

**Table 6 animals-14-03681-t006:** Post-absorptive plasma sulfur-containing metabolite concentrations (µmol/L) of primiparous sows ^1^.

	Dietary Treatment ^1^				SEM ^2^	*p*-Value ^3^		
Item	0.29		0.92					
	Farrow	Wean	Farrow	Wean		Diet	Stage	Diet × Stage
Homocysteine	25.34	12.28	23.82	13.16	0.88	0.711	<0.001	0.219
Cysteinyl–glycine	23.16	12.40	21.46	13.58	0.91	0.790	<0.001	0.084
Glutathione	1.406	0.397	1.813	0.452	0.65	0.741	0.098	0.791
Methionine	45.01	34.34	54.02	46.18	7.53	0.169	0.242	0.856
Cysteine	24.73	16.31	17.55	34.40	8.77	0.533	0.642	0.170
Taurine	107.6 ^b^	39.60 ^c^	155.8 ^a^	36.60 ^c^	7.31	0.002	<0.001	0.003

^1^ Experimental diets were fed between gestation day 90 and farrowing. Dietary treatments were as follows: 0.29% and 0.92% standardized ileal digestible sulfur AAs, which correspond to 63 or 200% of the estimated SID SAAs requirement for primiparous sows in late gestation (requirements from d 90 to d 114 of gestation, 145 kg of body weight at breeding, expected litter size of 15, and piglet birth weight of 1.35 kg). A common lactation diet was provided from farrowing through to weaning (23 ± 5 days of lactation). Blood samples were collected during the post-absorptive stage, 15 h after the last meal. ^2^ Maximum value for the standard error of the means. ^3^ *p*-values for the two-way ANOVA procedure between dietary treatment (0.29 vs. 0.92% standardized ileal digestible sulfur AAs,%), stage (farrowing and weaning), and the interaction. ^a,b,c^ means with different superscripts are statistically different within the stage.

**Table 7 animals-14-03681-t007:** Post-absorptive plasma sulfur-containing metabolite concentrations (µmol/L) of piglets at birth, weaning, and after three and six weeks of the nursery phase from sows fed diets containing 0.92 or 0.29% SID SAAs from day 90 to farrowing ^1^.

	Dietary Treatment ^1^								SEM ^4^	*p*-Value ^5^		
Item	0.29				0.92							
	Birth	Wean ^2^	Three Weeks ^3^	Six Weeks ^3^	Birth	Wean ^2^	Three Weeks ^3^	Six Weeks ^3^		Diet	Stage	Diet × Stage
Homocysteine	3.64 ^a^	36.71 ^b^	37.40 ^d^	25.69 ^e^	3.45 ^a^	51.62 ^c^	39.04 ^d^	28.82 ^e^	2.62	0.07	<0.001	<0.001
Cysteinyl–glycine	10.10	32.99	17.73	22.08	10.28	33.67	18.37	20.45	1.03	0.96	<0.001	0.595
Glutathione	1.078	0.813	1.653	1.484	0.801	1.128	2.01	0.995	0.28	0.89	0.105	0.197
Methionine	18.11	50.62	45.67	51.84	31.00	51.6	41.75	49.98	5.19	0.62	<0.001	0.321
Cysteine	43.22	27.73	23.05	40.12	44.49	32.24	22.65	28.72	6.36	0.73	0.017	0.564
Taurine	126.4	117.9	77.80	92.32	114.6	119.9	57.7	92.42	9.89	0.3	<0.001	0.699

^1^ Dietary treatments were as follows: 0.29% and 0.92% standardized ileal digestible sulfur AAs, which correspond to 63 or 200% of the estimated SID SAAs requirement for primiparous sows in late gestation (requirements from d 90 to d 114 of gestation, 145 kg of body weight at breeding, expected litter size of 15, and piglet birth weight of 1.35 kg). Common diets were provided in a 4-phase feeding program after weaning (Phase I: wean to d 7; Phase II: d 8 to d 21; Phase III: d 22 to d 35; Phase IV: d 36 to end of trial). Blood samples were collected after farrowing, within 1 h post-parturition before 1st suckling, at weaning, and in post-absorptive stage, 15 h after the last meal for weeks 3 and 6 of nursery. ^2^ Weaning occurred after 23 ± 5 days. ^3^ Analysis performed to evaluate a possible carry-over effect of experimental diets containing 0.29% or 0.92% standardized ileal digestible sulfur amino acids fed to the gilt between d 90 of gestation and farrowing. ^4^ Maximum value for the standard error of the means. ^5^ *p*-values for the two-way ANOVA procedure between the dietary treatments, stage (birth, wean, 3rd and 6th week of nursery), and the interaction. ^a,b,c,d,e^ means with different superscript are statistically different within the stage.

**Table 8 animals-14-03681-t008:** Post-absorptive plasma amino acid concentrations in gestating primiparous sows at farrowing and at weaning ^1^.

	Dietary Treatment ^1^				SEM ^2^	*p*-Value ^3^		
Item	0.29		0.92					
	Farrow	Wean	Farrow	Wean		Diet	Stage	Diet × Stage
Indispensable AA, μM								
Arg	166.0	103.5	169.8	100.0	10.45	0.988	<0.001	0.725
His	111.6	92.10	113.4	87.80	6.75	0.867	<0.001	0.628
Ile	128.8	139.4	152.7	153.5	9.70	0.049	0.583	0.633
Leu	169.6	196.6	206.6	211.3	11.52	0.042	0.161	0.321
Lys	192.6	134.9	185.7	133.7	11.51	0.730	<0.001	0.806
Met	45.01	34.34	54.02	46.18	7.53	0.169	0.242	0.856
Phe	125.5	96.50	136.3	111.1	8.89	0.144	0.007	0.842
Thr	334.8	137.2	312.6	133.6	21.20	0.534	<0.001	0.676
Trp	75.51	75.53	83.38	95.25	10.21	0.125	0.611	0.612
Val	284.1	237.6	364.6	247.6	18.36	0.018	<0.001	0.075
Total	1609	1248	1779	1320	86.74	0.166	<0.001	0.588
Dispensable AA, μM								
Ala	599.2	243.5	567.8	234.7	25.70	0.455	<0.001	0.655
Asn	75.43	54.37	83.59	58.01	5.11	0.210	<0.001	0.691
Asp	14.17	9.43	14.38	17.94	3.28	0.183	0.864	0.230
Gln	3835	2339	3573	2090	243.0	0.334	<0.001	0.978
Glu	144.1	96.6	146.5	93.7	6.17	0.976	<0.001	0.624
Gly	736	674	726	601	38.22	0.313	0.016	0.390
Pro	246.7	161.6	263.7	163.3	11.21	0.397	<0.001	0.515
Ser	124.7	123.9	179.5	122.9	13.55	0.071	0.035	0.040
Tyr	107.9	66.98	122.7	90.8	9.54	0.039	0.001	0.663
Total ^4^	6015	3825	5851	3543	274.4	0.446	<0.001	0.823

^1^ Dietary treatments were as follows: 0.29% and 0.92% standardized ileal digestible sulfur AAs, which correspond to 63 or 200% of the estimated SID SAAs requirement for primiparous sows in late gestation (requirements from d 90 to d 114 of gestation, 145 kg of body weight at breeding, expected litter size of 15, and piglet birth weight of 1.35 kg). A standard lactation diet was provided to all primiparous sows from farrowing through to weaning (23 ± 5 days of lactation); blood samples were collected at the post-absorptive stage, 15 h after the last meal. ^2^ Maximum value for the standard error of the means. ^3^ *p*-values for the two-way ANOVA procedure between dietary treatments, stage (farrowing and weaning), and the interaction. ^4^ Cys was included in the total.

**Table 9 animals-14-03681-t009:** Post-absorptive plasma AA concentrations (µmol/L) of piglets at birth, weaning, and after three and six weeks of the nursery phase ^1^.

	Dietary Treatment ^1^								SEM ^4^	*p*-Value ^5^		
Item	0.29				0.92							
	Birth	Wean ^2^	3 Weeks ^3^	6 Weeks ^3^	Birth	Wean ^2^	3 Weeks ^3^	6 Weeks ^3^		Diet	Stage	Diet × Stage
Indispensable AA, μM ^6^												
Arg	112.0	114.8	161	169.7	73.6	131.1	140.8	159.5	19.32	0.367	<0.001	0.455
His	53.9	78.9	64.1	61.6	98.4	85.9	62.4	66.33	20.54	0.299	0.749	0.662
Ile	54.9	120	190.5	216.4	55.9	130.2	189.4	229.2	8.81	0.389	<0.001	0.825
Leu	101.6	152.2	216.6	242.3	122.5	153.2	220.7	260.3	15.90	0.388	<0.001	0.879
Lys	174.7	146.5	161.3	208.5	186.4	142.8	161.2	224.4	15.30	0.609	<0.001	0.890
Met	18.11	50.62	45.67	51.84	31	51.6	41.75	49.98	5.19	0.619	<0.001	0.321
Phe	67.2	121.1	116.7	117.2	82.1	128.8	114.1	116.1	7.33	0.432	<0.001	0.563
Thr	85.6	103.8	161.5	166.4	92.1	114.4	150.4	181.9	11.19	0.523	<0.001	0.748
Trp	42.23	66.79	62.42	71.05	41.09	75.72	51.32	70.74	4.00	0.802	<0.001	0.187
Val	233.5	228.3	313.1	366.3	300.1	234.2	297.9	393.5	17.11	0.128	<0.001	0.099
Total	944	1183	1493	1671	1083	1248	1430	1736	88.43	0.416	<0.001	0.784
Dispensable AA, μM												
Ala	741	301	312	318	668	317	275	362	38.00	0.608	<0.001	0.358
Asn	58.81	53.09	70.39	82.41	51.38	50.75	65.73	73.58	6.39	0.201	<0.001	0.952
Asp	22.26	13.39	15.78	19.74	19.54	12.95	18.04	23.26	2.74	0.744	0.005	0.624
Cys	43.22	27.73	23.05	40.12	44.49	32.24	22.65	28.72	6.36	0.725	0.017	0.564
Gln	2795	2176	3077	3425	3434	2100	2808	3552	316.00	0.658	<0.001	0.500
Glu	168.9	130.2	165.9	166.1	170.1	131.3	165.5	177.2	15.43	0.77	0.016	0.981
Gly	923	1054	754	850	1146	1028	825	866	188.00	0.63	0.445	0.929
Pro	222.7	229.1	196.6	241.3	253.7	224.4	190.8	253.9	24.77	0.748	0.201	0.799
Ser	231.3	155.7	188.1	187.9	204.8	162.4	205.8	202.8	11.61	0.648	<0.001	0.186
Tyr	79.7	111.8	98.1	130.6	119.6	127.5	91.6	129	13.93	0.482	0.023	0.163
Total ^7^	5412	4370	4979	5554	6226	4306	4725	5761	441	0.54	0.001	0.652

^1^ Dietary treatments were as follows: 0.29% and 0.92% standardized ileal digestible sulfur AAs, which correspond to 63 or 200% of the estimated SID SAAs requirement for primiparous sows in late gestation (requirements from d 90 to d 114 of gestation, 145 kg of body weight at breeding, expected litter size of 15, and piglet birth weight of 1.35 kg). Common diets were provided in a 4-phase feeding program (Phase I: wean to d 7; Phase II: d 8 to d 21; Phase III: d 22 to d 35; Phase IV: d 36 to end of trial). ^2^ While sows were receiving standard lactation diets for 23 ± 5 days. ^3^ Analysis performed to evaluate a possible carry-over effect of experimental diets containing 0.29% or 0.92% standardized ileal digestible SA fed between d 90 of gestation and farrowing on the offspring during nursery stage. ^4^ Maximum value for the standard error of the means. ^5^ *p*-values for the two-way ANOVA procedure between the treatments and stage (farrowing, weaning, 3rd and 6th week of nursery). ^6^ Blood samples were collected after farrowing, within 1 h post-parturition before 1st suckling, at weaning, and in post-absorptive stage, 15 h after the last meal for weeks 3 and 6 of nursery. ^7^ Cys was included in the total.

## Data Availability

The raw data supporting the conclusions of this article will be made available by the authors on request.

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
