# Peer review of "Effect of Low- and High-Sulfur-Containing Amino Acid Inclusion in Diets Fed to Primiparous Sows in Late Gestation on Pre-Partum Nitrogen Retention and Offspring Pre- and Post-Weaning Growth Performance"

_animals, 2024, doi:10.3390/ani14243681_

Round 1

Reviewer 1 Report

Comments and Suggestions for Authors

I think this is good information, we definitely need to have more trials conducted to understand the role of amino acids in sows, and especially methionine/SAA. I do encourage the author (s) to elaborate more on the materials and methods so the next person that wants to continue this line of investigation is able to replicate your results. Thank you for putting this together

Author Response

Animals ISSN 2076-2615 Manuscript ID animals-3382549 Title

Effect of low and high sulfur-containing amino acid inclusion in diets fed to primiparous sows in late gestation on pre-partum nitrogen retention and offspring pre- and post-weaning growth performance

Authors Cristhiam Jhoseph Munoz Alfonso, Lee-Anne Huber, Crystal L. Levesque * Section Pigs

Special Issue Maternal Nutrition and Neonatal Development of Pig

Line 14= remove or reword “some plasma sulfur-containing metabolites of the offspring” either

clarify which metabolites or just use “growth performance of the offspring”

R= Metabolites were included.

Line 26= don’t you need to add “±” on d 89 of gestation? And also, should you say sows were “selected” since in line 32 you mention gilts received the assigned diet between day 90 of gestation and farrowing. Or make it consistent and use either day 89 or 90.

R= Thanks for your comment. I see how it can be misunderstood. However, the selection was done at the theoretical “exact d 89 of gestation” using the breeding day as reference. Also, the sows were selected that day and the following day (d 90) the corresponding experimental diet was provided for the first time.

Line 32= add “late gestation” to the entire document to make it consistent with the title of the article. These diets were only evaluated during the last 20-25 days. In addition, add a period after farrowing. Then, explain the sows received a standard lactation diet after farrowing and pigs received a standard commercial diet after weaning.

R= late gestation included everywhere when days are not stated (to avoid redundancy).

Line 40= remove word “also”

R= done

Line 41= need to define Tau, I’m assuming is taurine. If so, need to define this in line 17

R= done

Line 51= please add “balance” instead of “balanced”

R= done

Line 53= please add “improve” rather than “improved”

R= done

Line 61= remove “themselves” and end sentence there

R= done

Line 63= remove “s” in product

R= done

Line 65= end sentence in “maternal tissues”

R= done

Line 87= in the abstract section 27 sows are mentioned, in line 87 includes 30 sows. If sows were lost during the experiment, need to report how many were removed and why, and add something like “only the results of 27 sows were included and will be discussed”

R= Thanks for your comment. This suggested change was included at the beginning of the results. The experiment was designed to have equal number of replicates (15) but the reasons for excluding the sows are included down below.

Line 93-95= I think I know what you mean when you say that up to 8 piglets/litter were placed in nursery pens, 1 litter per pen. However, I think this can be worded more clearly. Something along the lines of “litters from sows receiving the dietary treatments remained together (up to 8

pigs/litter) after weaning. Placing 4 females and 4 males per pen, for six weeks”

R= Thanks for the suggestion, this edit has been included in the manuscript

Line 97= Use corn, then peas, rye etc..

R= done

Line 98= should you include the 65 as your control? The NRC recommends 0.29% so that would indicate the 65% SID SAA would be considered your control treatment

R=Thank you for comment. The NRC recommendation is 0.46 % SID SAA using the specifications described in section 2.2. Therefore, 0.29 and 0.92 % SID SAA are the theoretical ~60 % and ~200 % inclusions respectively.

Line 100= how the expected litter size and piglet birth weight was estimated? By using farm’s

previous primiparous sows performance? Please define or delete sentence

R= The parameter were taken based on the expected values provided by the genetic company and local records.

Line 107= the experimental diets (2.50kg/day) is different than the stated in line 32 (2.53kg) please choose one and correct for the entire document

R= the correct value is 2.50 kg/day

Line 110= please explain/define phase duration of the nursery program like you did in line 240

R= done

Line 125= suggest changing to “fed to sows during late gestation from day 90 to farrowing”. Also,

you have 60% here where in line 98 and in line 30 you have 65% and 60%, which one is it?

R= You are right. The actual value os 63%. All values have been adjusted accordingly.

Line 133= remove what is in parentheses

R= done

Line 135= should you include the brand of urinary catheters used? And how the daily fecal grab sampling was performed? This is a scientific paper; the reader should be able to replicate this following your materials and methods. Feel free to expand in text.

R= Thanks for your comment. Some information has been added as requested in this section, However, the entire procedure has been described previously and it is allowed to include the citation if the procedure is not altered or modified.

Line 139= were all the sows used to collect blood? Or did you select a few? If so, please clarify

R= all sows were sampled. Info included in the paper.

Line 141= I would add more detail. To collect blood samples from sows did you use a snare?

R= yes, that was the method used to restrain them.

Line 150= I’m curious how you will discuss this in your results. If different pigs were sampled, how can you correlate if there were any changes due to treatment? This increase variability. While this method is not incorrect, it would be great if you can explain why this sampling method was chosen.

R= I understand why you are curious about this topic. While the experimental unit during gestation and lactation is the sow. At nursery the experimental unit is the pen. Therefore, we selected the animals based on an average of the individuals included in the experimental unit. This allow us to describe the carry-over effect that the maternal supplementation can have over the offspring, and not only over a pig that can also die during the nursery period increasing the number of missing samples.

Line 154= define Met here and elsewhere. I know it is referring to methionine, but all definitions should be included

R= done

Line 172= add “respectively” after cysteine acid. And define Cys here and elsewhere.

R= done

Line 177= if you are using capital letters n line 180, should you not use capital letters here?

R=Glutathione has been modified as requested

Line 183= briefly describe the minor adaptations

R= more information regarding this comment  was included in the revision.

Line 190- “T-Test” instead of TTest

R= done

Line 194= the blood samples from pigs should be analyzed or reported differently than “repeated measures” this does not apply to pigs as you didn’t use the same pig to collect blood from in the first place. I’m assuming you did use the same sow to collect blood samples and that would be a repeated measures

R= as described above, the experimental unit is the pen. Therefore, the repeated measurement is allowed here

Line 202= please refer to my quoted comment in line 87.

R= Based on formatting advice, here was the best place to describe these events.

Line 217= move table 3 to the next page. Also, why is no SEM or P-value reported for N intake?

R= Feed refusal was not predominant over the course of the experiment; for the few cases where feed was not consumed completely, the data were excluded from N balance calculations. Since there was only one batch of diets mixed, each treatment had a single value for the N intake and thus, standard error was not calculated.This information has been included in the manuscript

Line 230= define on Table 4 how the sow BW and backfat change was done. The overall average

daily gain is applicable to the litter I presume, if so, please add “litter overall average daily gain, kg”

D= done

Line 269= Table 6 and Table 7 have uneven bolded letters, please correct

Done

Line 288= you can not call it a carry-over effect since you did not sample the same pig overtime.

R= experimental unit during the nursery period was the pen

Line 295= I’m in the fence of reporting only the main effects of stage and not the contrast P-values, can you please explain what additional information the contrast P-values provide? What are the contrasts? That is not defined in the document

R= Thanks for your comment. The main effect of diet does the comparison independently of stage, stage do not take into account treatment, the interactions show that at certain stage, the treatment had an effect. Therefore, the contrast by stage helps to elucidate where the treatment had an effect within each stage. (a,b,c superscript describes this comparison)

Line 315= remove “this”

Done

Line 319= please refer to comment on line 125

R= tables were corrected

Line 326= why was included in the total and not as individual? It would be good if you can explain

R= This clarification was to guarantee to the reader that the total includes  Cys as its analysis was included in other table (Met metabolites).

Line 333= please refer to comment on line 194

R= this comment has been addressed.

Line 395= do you need to have “Tau” before your reference #26?

R= this comment has been addressed.

Line 402= should we add “such as”?

R= thank you for the suggestion. However, we considered no needed to improve clarity of the sentence.

Line 435= remove “AA such” and just use Leucine instead. And might be good to add what other

amino acids are being boosted by circulating levels of Leucine.

R= correction accepted. However, no more information has been included about AAs because the main purpose of the sentence was to highlight the effect of Tau on m-TOR which may influence all AAs used for protein synthesis depending of the type of protein.

Reviewer 2 Report

Comments and Suggestions for Authors

Dear Authors

The manuscript is clear and very interesing. The experimental desing is appropriate. The conclusions are presented well and align with the results obtained during the investigation.

A few comment to the text:

L86: Please provide information on:

- location of the farm, its population and broadly understood production standards

- health status of the sows and prophylaxis potentially affecting the number of piglets born alive or mummified

- age at the first insemination

L91: Did you administer antibiotics perinatally? What about piglet processing?

L96: Can you please add information on the type of mill, steam conditioning and pelleting process?

L113-121: Please see the formatting guide and correct the captions. The comment refers to all the tables in your manuscript.

L255: Please correct the error (= <).

L258, 261: Please correct the spacing errors (P =0.xxx).

L269: Please remove the line below Farrow. Also, please remove the bold print.

L271: Please correct the spacing error (200%) and double check the entire manuscripts (e.g. L281, 288).

L279: Please remove the bold print (third line: Birth, etc.).

L312: total, not Total

L317: Please remove the bold print (Farrow, 0.29). Also please see Table 9 and correct the same errors.

L383, 401: Please add the names.

L390-391: Please avoid the repetition (grater).

L396-398: Does the speculation refer to [26] or other pieces of research?

L449: Please consider replacing the word outstanding.

best regards

Author Response

The manuscript is clear and very interesing. The experimental desing is appropriate. The conclusions are presented well and align with the results obtained during the investigation.

Dear reviewer: Thanks for your comments and the time spent on them. All your comments have been addressed in the final version of the manuscript. Below you will find comments on each of your suggestions as required by the journal.

A few comment to the text:

L86: Please provide information on:

- location of the farm, its population and broadly understood production standards

R= The present study was conducted at the South Dakota State University Swine Education and Research facility.

- health status of the sows and prophylaxis potentially affecting the number of piglets born alive or mummified

R= I understand the nature of your comment. However, it comes from a research facility where all experiments are evaluated and monitored by a veterinary committee. The health status is checked to continuing the experiment.

- age at the first insemination

R= Primiparous sows were bred at day 470 ± 5 of life.

L91: Did you administer antibiotics perinatally? What about piglet processing?
R= it was not needed to supply antibiotics for the sows at any point of the experiment. For the litter, standard litter processing procedures were performed such as castration and tooth clipping.

L96: Can you please add information on the type of mill, steam conditioning and pelleting process?

R= Diets were provided in mash form.

L113-121: Please see the formatting guide and correct the captions. The comment refers to all the tables in your manuscript.

R= subtitles were corrected and formatted as well as the tables using one published example.

L255: Please correct the error (= <).

R= Done

L258, 261: Please correct the spacing errors (P =0.xxx).

R= Done

L269: Please remove the line below Farrow. Also, please remove the bold print.

R= Done

L271: Please correct the spacing error (200%) and double check the entire manuscripts (e.g. L281, 288).

R= Thanks for the observation. All %’s icons have been verified.

L279: Please remove the bold print (third line: Birth, etc.).

R= Done

L312: total, not Total

R= Done

L317: Please remove the bold print (Farrow, 0.29). Also please see Table 9 and correct the same errors.
R= Done

L383, 401: Please add the names.

R= correct citation included in the text

L390-391: Please avoid the repetition (grater).

R=done

L396-398: Does the speculation refer to [26] or other pieces of research?

R=the reference is for the fact stated right before the quotation, the speculation is done because no reference was found stating otherwise. However, more information was included in the sentence to clarify.

L449: Please consider replacing the word outstanding.

R=done
